# 2′-Fucosyllactose and 3-Fucosyllactose Alleviates Interleukin-6-Induced Barrier Dysfunction and Dextran Sodium Sulfate-Induced Colitis by Improving Intestinal Barrier Function and Modulating the Intestinal Microbiome

**DOI:** 10.3390/nu15081845

**Published:** 2023-04-12

**Authors:** Yeon-Ji Kim, Han-Hae Kim, Chul-Soo Shin, Jong-Won Yoon, Seon-Min Jeon, Young-Ha Song, Kwang-Youn Kim, Kyungho Kim

**Affiliations:** 1Korean Medicine (KM)—Application Center, Korea Institute of Oriental Medicine (KIOM), Daegu 41062, Republic of Korea; yjikim@kiom.re.kr (Y.-J.K.); kdh948@kiom.re.kr (H.-H.K.); 2Advanced Protein Technologies Corp., Suwon 16229, Republic of Korea; csshin@aptech.biz (C.-S.S.); jwyoon@aptech.biz (J.-W.Y.); smjeon@aptech.biz (S.-M.J.); yhsong@aptech.biz (Y.-H.S.)

**Keywords:** 2′-fucosyllactose, 3-fucosyllactose, inflammation, tight junction, ulcerative colitis

## Abstract

Ulcerative colitis is an inflammatory bowel disease (IBD) with relapsing and remitting patterns, and it is caused by varied factors, such as the intestinal inflammation extent and duration. We examined the preventative effects of human milk oligosaccharides (HMOs) on epithelial barrier integrity and intestinal inflammation in an interleukin (IL)-6-induced cell model and dextran sodium sulfate (DSS)-induced acute mouse colitis model. HMOs including 2′-fucosyllactose (FL) and 3-FL and positive controls including fructooligosaccharide (FOS) and 5-acetylsalicylic acid (5-ASA) were orally administrated once per day to C57BL/6J mice with colitis induced by 5% DSS in the administered drinking water. 2′-FL and 3-FL did not affect the cell viability in Caco-2 cells. Meanwhile, these agents reversed IL-6-reduced intestinal barrier function in Caco-2 cells. Furthermore, 2′-FL and 3-FL reversed the body weight loss and the remarkably short colon lengths in DSS-induced acute colitis mice. Moreover, 2′-FL and 3-FL obviously protected the decreasing expression of zonula occluden-1 and occludin in colon tissue relative to the findings in the DSS-treated control group. 2′-FL and 3-FL significantly reduced IL-6 and tumor necrosis factor-α levels in serum relative to the control findings. The summary of these results shows that HMOs prevent colitis mainly by enhancing intestinal barrier function and advancing anti-inflammatory responses. Therefore, HMOs might suppress inflammatory responses and represent candidate treatments for IBD that protect intestinal integrity.

## 1. Introduction

Inflammatory bowel disease (IBD) is a chronic and incurable disease that causes inflammation and ulcers of unknown cause in the digestive tract. IBD can be categorized as ulcerative colitis (UC) or Crohn’s disease (CD) [1,2]. The worldwide number of patients with IBD is estimated to exceed six million, and the incidence rates of UC and CD in the Americas/Europe are estimated as 24.3 per 100,000 and 29.3 per 100,000, respectively [3]. Among Asian countries, the incidence of IBD is rapidly increasing in Westernized/advanced countries including South Korea [4]. Intestinal mucous membranes and intestinal microbes interact with substances introduced exogenously such as food and pathogens, and these interactions play a critical function in supporting the human organism from various external stimuli and maintaining intestinal motility [5]. Intestinal bacterial translocation is dependent on the permeability of intestinal epithelial cells, which is controlled by tight junctions (TJs) [6]. TJs can influence the intestinal permeation of substances by regulating intercellular niches [7]. When TJs are loosened because of intestinal stimulation or damage, harmful molecules (e.g., pathogens, toxins, antigens) excessively accumulate in the intestinal epithelium, causing endotoxemia, mucosal immune system disruption and inflammatory responses, ultimately leading to intestinal or systemic diseases [8]. Recent studies revealed that enhancing the protective effects of TJ integrity and reducing the inflammatory response are strategies for IBD treatment [9]. Accumulating evidence indicates that TJs have important roles in gut permeability for the maintenance of intestinal completeness, metabolic and immunological response [7,10]. TJs are composed of tough junctional complexes such as transmembrane adhesion proteins (e.g., claudins, occludin), intracellular adaptor proteins (e.g., zonula occludens [ZO]-1, ZO-2, ZO-3), and the actin cytoskeleton, and collectively constitute a physical barrier to protect the luminal microbiota [11,12].

Human milk oligosaccharides (HMOs) of breast milk exist in a significantly higher concentration (15 g/L) than their counterparts in other mammals, and more than 200 of such oligosaccharides have been identified [13,14]. Regarding the main function of HMOs, research related to their prebiotic effect that promotes the growth of intestinal lactic acid bacteria has been recently reported [13]. HMOs consist of five monosaccharides: D-galactose, D-glucose, N-acetylglucosamine, L-fucose and sialic acid (N-acetylneuraminic acid) [15,16]. In spite of their structural complexity, HMOs have several common structures [13]. Most HMOs have a lactose moiety at the reducing end. The galactose moiety of lactose is sialylated to form 3′-sialyllactose (SL) and 6′-SL through α-(2,3)- and α-(2,6)-bonds, respectively, whereas α-(1,2)- and α-(1,3)-bonds can be fucosylated to form 2′-fucosyllactose (FL) and 3-FL, respectively [15]. FL is the major constituent of HMOs and 2′-FL. The most abundant HMO in human milk is 2′-FL, accounting for 20% of the total HMO content [17]. Meanwhile, 3-FL, the concentration of which ranges from 0.3 to 0.58 g/L, accounts for less than 1% of all HMOs [18,19]. Preclinical studies have demonstrated that 2′-FL has diverse biological effects affecting bacteria–epithelial cell interactions, leukocyte adhesion and neuronal development [20]. Moreover, 2′-FL primarily protects against infection and inflammation [13,21,22]. The prebiotic effects of HMOs have been extensively studied. HMOs in breast milk selectively promote the growth of Bifidobacterium bifidum, which is dominant in the early intestine of infants, but does not affect the growth of harmful bacteria [14]. In addition, HMOs inhibit the intestinal adhesion of pathogens because the cell wall polysaccharide structure of pathogens that binds intestinal lectin is often similar to some HMO structures [23,24]. Although there is increasing evidence for the beneficial effects of 2′-FL and 3-FL, their potential benefit in IBD is unclear. Therefore, in this study, the protective effects of 2′-FL and 3-FL against dextran sodium sulfate (DSS)-induced colitis were investigated. This study revealed that both 2′-FL and 3-FL decreased intestinal permeability in vivo and in vitro. As the final outcome, these compounds consequently prevent the reduction in TJ protein expression following exposure to interleukin (IL)-6 or DSS and diminish inflammation. Hence, these saccharides could be effective therapeutic agents for IBD.

## 2. Materials and Methods

### 2.1. Chemicals and Reagents

2′-FL and 3-FL were obtained from Advanced Protein Technologies Corp. (Suwon, Republic of Korea). Fructooligosaccharide (FOS) was purchased from Neo Cremar (Seoul, Republic of Korea). 5-aminosalicylic acid (5-ASA), Dextran sodium sulfate (DSS) and hematoxylin and eosin (H&E) solutions were purchased from Sigma Aldrich (St Louis, MO, USA) and MP Biomedicals (Santa Ana, CA, USA), respectively. Enzyme-linked immunosorbent assay (ELISA) kits for mouse interleukin (IL)-6 and tumor necrosis factor (TNF)-α and myeloperoxidase (MPO) activity were purchased from eBioscience (San Diego, CA, USA) and Thermo Fisher Scientific (Waltham, MA, USA), respectively. RIPA lysis buffer and phosphatase and protease inhibitor cocktails were obtained from Millipore (Darmstadt, Germany) and Roche (Basel, Switzerland), respectively. Anti-ZO-1, anti-occludin and anti-claudin-2 antibodies were purchased from Thermo Fisher Scientific. 5-ASA was used as positive control.

### 2.2. Cell Culture and Viability Assay

Caco-2 cells (ATCC, Manassas, VA, USA) were cultured in DMEM (HyClone, Logan, UT, USA) supplemented with 1% antibiotics (Hy-Clone) at 37 °C in an atmosphere of 5% CO_2_. Caco-2 cells were seeded at 1 × 10^5^ cells/well and pre-treated with various concentrations of 2′-FL, 3-FL and FOS for 1 h, following an additional 24 h of treatment with 50 ng/mL IL-6. Cell viability was evaluated using Cell Counting Kit-8 (Dojindo Molecular Technologies Inc., Rockville, MD, USA) and absorbance at 450 nm using a VersaMax microplate reader (Molecular Devices, Sunnyvale, CA, USA). Cell viability (% control) was calculated relative to that 100× absorbance of treated sample/absorbance of vehicle [9].

### 2.3. Measurement of Transepithelial Electrical Resistance (TEER) and Epithelial Paracellular Permeability

Caco-2 cells were seeded at 1 × 10^5^ cells/insert in Polyethylene Terephthalate hanging cell culture insert with pore size of 0.4 µm in a 24-well plate (Millipore, Bedford, MA, USA). The medium was changed every 2~3 days until complete differentiation during 21 days. Cells were pretreatment with various concentrations of 2′-FL, 3-FL and FOS for 1 h and then treated with 50 ng/mL IL-6. The electrical resistance was measured in three independent measurements using a Millicell ERS-2 voltohmmeter (Millipore). TEER were determined after treatment for 18 and 24 h and presented as Ohm·cm^2^. Paracellular permeability was determined using a nonabsorbable FITC-conjugated dextran probe (FD-4). After a 24 h treatment with IL-6 and/or 2′-FL, 3-FL and FOS, 1 mg/mL FD-4 was added on the apical side, and PBS was added on the basolateral side and then incubated for 1 h at 37 °C. Absorbance of basolateral side was calculated at excitation and emission wavelengths of 490 and 520 nm, respectively, using a VersaMax microplate reader [9].

### 2.4. Mice and Treatment

All experimental animal procedures were approved by the Institutional Animal Care and Use Committee of Korea Institute of Oriental Medicine (KIOM-21-081) and performed in accordance with their guidelines. Six-week-old male C57/BL6 mice were purchased from DooYeol Biotech (Seoul, Republic of Korea) and divided into nine groups: vehicle-treated control (*n* = 10), 5% DSS (*n* = 10), 5% DSS + 2′-FL Low (*n* = 10), 5% DSS + 2′-FL High (*n* = 10), 5% DSS + 3-FL Low (*n* = 10), 5% DSS + 3-FL High (*n* = 10), 5% DSS + FOS Low (*n* = 10), 5% DSS + FOS High (*n* = 10), and 5% DSS + 100 mg/kg 5-ASA (*n* = 10). DSS-induced colitis mice were established by feeding mice 5% (wt/vol) DSS (MP Biomedicals, molecular weight of 36,000–50,000) dissolved in drinking water for 5 days, followed by 3 days of DSS-free water drinking [9]. Starting from the first day of DSS challenge, 2′-FL, 3-FL, FOS or 5-ASA was orally administered at the indicated dose per day, and weight was measured daily before administration. After the end of the experiment and sacrifice, colon length was measured and photographed.

### 2.5. Evaluation of the Disease Activity Index (DAI)

The body weight, stool condition, and gross bleeding of the mice were recorded daily to assess the severity during the experimental schedule (Table 1). DAI was determined as an average of the scores for [weight loss + stool condition + gross bleeding]/3 [25].

### 2.6. Quantification of In Vivo Epithelial Paracellular Permeability

On day 8 of the experiment, we investigated paracellular epithelial permeability using FD-4. The mice were fed 60 mg/100 g FD-4 solution by gavage. After 4 h, serum was collected, and the FD-4 content was measured at 490 nm excitation and 520 nm emission using a VERSAmax microplate reader [26].

### 2.7. Large-Intestine Endoscopy and Histological Analysis

We obtained the higher-resolution images of anesthetized mice colons with isoflurane using a mini-endoscope with a visible light source (OLYMPUS, Tokyo, Japan; 670 mm length and 2.8 mm diameter) on day 8 of the experiment. After endoscopy, mice were sacrificed; subsequently, whole blood and intestinal tissue were collected. Then, we performed histopathological analysis using colon tissue sections stained with H&E solution [27].

### 2.8. ELISA for MPO Activity, IL-6, and TNF-α

The IL-6 and TNF-α levels were determined in serum using ELISA kits according to the manufacturer’s protocol. In addition, MPO activities were determined in homogenized colon tissue using an MPO activity assay kit according to the manufacturer’s protocol [9].

### 2.9. Western Blot Analysis

Equal amounts of cell and mouse colon proteins were separated on SDS-PAGE gels and then incubated with primary and secondary antibody and examined using an Alliance Q9 Advanced Chemiluminescence Imager (Cleaver Scientific Ltd., Warwickshire, UK). The band density was then normalized to that of β-actin as the reference [27].

### 2.10. Intestinal Microbiome Analysis

DSS was utilized to induce colitis in 23 mice. Of these, eight mice were administered a high dose of FOS (High; *n* = 3), a low dose of FOS (Low; *n* = 2), or 5-ASA (*n* = 3) as positive controls. The remaining mice with colitis, excluding three disease controls, were administered a high dose of 2′-FL (2′-FL-High; *n* = 3), low dose of 2′-FL (2′-FL-Low; *n* = 3), a high dose of 3-FL (3-FL-High; *n* = 3), and low dose of 3-FL (3-FL-Low; *n* = 3) as experimental models. Feces from each group were collected to examine the effect of HMO administration on the gut microbiome under colitis. We obtained the gut microbiome from mouse feces and analyzed the microbes using the shotgun metagenome sequencing method with high resolution. Using a mouse gut metagenome catalog (MGS catalog; http://dx.doi.org/10.5524/100114, accessed on 3 May 2022) and METEOR software (http://www.jobim2010.fr/sites/default/files/presentations/27Pons.pdf, accessed on 3 May 2022), we detected the gut microbiome in mouse feces after removing chimeric sequences from different species. Then, the abundance of *Escherichia coli* (*E. coli*), alpha and beta microbial diversity, the bacterial composition, and the main species of the gut microbiome were compared between the groups using the R packages “microbiome” (http://microbiome.github.com/microbiome, accessed on 3 May 2022) and “MOMR” [28].

### 2.11. Statistical Analysis

All graphs were drawn with GraphPad Prism version 5 and indicated as the means ± standard error of the mean. The significant difference was determined at *p*-value < 0.05. Post hoc comparison of means was carried out with the Tukey’s test (one-way ANOVA) or with the Bonferroni’s test (two-way ANOVA) for multiple comparisons when appropriate.

## 3. Results

### 3.1. 2′-FL and 3-FL Protect against IL-6-Induced Epithelial Barrier Dysfunction in Caco-2 Cells

First, we determined the effects of 2′-FL and 3-FL on the viability of Caco-2 cells. Results showed that 2′-FL and 3′-FL alone did not affect the viability of Caco-2 cells (Figure 1A), and similar results were obtained for combined treatment with IL-6 (Figure 1A). Next, we measured TEER and FITC-dextran flux to evaluate epithelial barrier dysfunction. Our previous study observed increasing epithelial paracellular permeability with a maximal decrease in TEER after treatment with 50 ng/mL IL-6 for 24 h [14]. In this study, pretreatment with 2′-FL and 3-FL significantly increased TEER in a time- and concentration-dependent manner compared to the effect of treatment with IL-6 alone (Figure 1B,C). Furthermore, pretreatment with 2′-FL and 3-FL decreased FD-4 permeability in a concentration-dependent manner compared to the effect of IL-6 treatment alone (Figure 1D). 

### 3.2. 2′-FL and 3-FL Recover the TJ Protein Expression in IL-6-Stimulated Caco-2 Cells

To evaluate the effects of 2′-FL and 3-FL on intestinal mucosal barrier dysfunction, we confirmed the expression of TJ proteins, such as ZO-1, occludin and claudin-2, in Caco-2 cells. Pretreatment with 2′-FL and 3-FL reversed the IL-6-induced downregulation of ZO-1 and occludin (Figure 2A–C). Furthermore, 2′-FL and 3-FL decreased the IL-6-induced upregulation of claudin-2 (Figure 2A,D). Of these, 2′-FL more strongly reversed IL-6-induced change in ZO-1, occludin, and claudin-2 expression than 3-FL (Figure 2). These results indicated that 2′-FL obviously reverses IL-6-mediated barrier dysfunction.

### 3.3. 2′-FL and 3-FL Reverse DSS-Induced Change in Body Weight, Colon Length, and Intestinal Permeability in Mice with Colitis

We assessed the protective effects of 2′-FL and 3-FL on the symptoms of DSS-induced colitis. Compared to the findings in the normal group, DSS-administered mice exhibited obviously reduced body weight (Figure 3A,B). Meanwhile, 2′-FL and 3-FL treatment significantly reversed body weight losses induced by DSS (Figure 3A,B). Furthermore, consistent with the change in body weight, DAI was significantly decreased by 2′-FL and 3-FL treatments compared to the effects of DSS administration alone (Figure 3C). In addition, DSS-induced decreases in colon length were remarkaby improved by 2′-FL and 3-FL treatment (Figure 3D,E). Compared to the findings in the normal group, serum FITC-dextran content was remarkably increased by 1.47 ± 0.08-fold in the DSS administration group, whereas FITC-dextran content was decreased by 0.83 ± 0.07, 0.82 ± 0.11, 0.84 ± 0.15-fold in the 2′-FL and 3-FL treatment groups, respectively (Figure 3F). These results suggest that HMOs treatment can significantly reverse increase in intestinal permeability induced by UC.

### 3.4. 2′-FL and 3-FL Attenuate the Histopathological Changes of Colon Tissues in Mice with Colitis

To assess the effects of 2′-FL and 3-FL on the histopathological and morphological changes in colon tissues induced by DSS treatment, we performed endoscopy and H&E staining. In the control group, epithelial cells were orderly arranged, and no epithelial injury and inflammatory cell infiltration were observed (Figure 4, H&E). The DSS group exhibited significant decrease in the height and thickness of the colonic villi and increased inflammatory cell infiltration on the mucous membrane (Figure 4, H&E). Meanwhile, in the 2′-FL and 3-FL groups, the average colon villus height and thickness were increased, and the number of infiltrating inflammatory cells on the mucous membrane was decreased (Figure 4, H&E). Moreover, compared to the effects of 3-FL and FOS, 2′-FL more strongly reduced histopathological damage. These results indicated that oral 2′-FL and 3′-FL can prevent DSS-induced colonic tissue damage.

### 3.5. 2′-FL and 3′-FL Alleviate DSS-Induced Intestinal Barrier Dysfunction in Mice

We evaluated the effects of 2′-FL and 3′-FL on TJ protein expressions in the tissues of mice with DSS-induced colitis. DSS administration resulted in significantly decreased ZO-1 and occludin expression in colon tissue (Figure 5A–C), whereas claudin-2 expression was increased (Figure 5A,D). However, these changes in protein expression were reversed by 2′-FL and 3-FL administration (Figure 5). Moreover, consistent with the Western blot results in the cell model, 2′-FL more strongly reversed changes in TJ protein expression than 3-FL (Figure 5). Taken together, these results indicated that 2′-FL and 3-FL protected against acute colitis by regulating TJ protein expression, and furthermore, 2′-FL could be used to prevent barrier dysfunction.

### 3.6. 2′-FL and 3-FL Alleviate MPO Activity and Changes of Pro-Inflammatory Cytokines Levels

In addition, to determine the effects of 2′-FL and 3-FL on DSS-induced MPO-mediated oxidative injury, we measured the chlorination and peroxidation activities of MPO using colon tissues. The findings demonstrated that DSS-administered mice had significantly higher chlorination and peroxidation activities than those in the control group (Figure 6A,B). Meanwhile, these effects of DSS were clearly reduced in the 2′-FL and 3-FLtreatment groups (Figure 6A,B). These results demonstrated that 2′-FL and 3-FL could reduce oxidative stress by regulating the concentrations of chloride and the reducing substrate. Furthermore, we measured the serum levels of IL-6 and TNF-α, which were significantly higher in the DSS-administered mice than in the control mice (Figure 7A,B). However, the serum levels of IL-6 and TNF-α were lower in mice treated with 2′-FL, 3-FL than in those treated with DSS alone (Figure 7A,B).

### 3.7. Effects of 2′-FL and 3-FL on the Intestinal Microbial Composition in Mice with DSS-Induced Colitis

The differences in microbiome biodiversity among the groups were determined by measuring alpha and beta diversity. Alpha diversity reflects microbial richness or abundance, whereas beta diversity indicates compositional differences between groups. We calculated Shannon’s entropy index to estimate alpha diversity, as it is considered to reflect the healthiness of the host (Figure 8A). We noticed that the value was highest in the healthy control group (median: 3.31), followed by the 5-ASA (median: 3.15), 2′-FL-High (median: 2.97), 3-FL-Low (median: 2.92), FOS Low (median: 2.90), 2′-FL-Low (median: 2.81), 3-FL-High (median: 2.53), FOS High (median: 2.49), and DSS control groups (median: 2.45). Shannon’s entropy index was significantly higher in the healthy control group than in all other groups excluding 5-ASA and 3-FL-Low groups. Surprisingly, the value was significantly higher in the 2′-FL-High group than in the DSS control groups. Beta diversity was estimated using principal coordinate analysis (PCoA) based on Bray–Curtis dissimilarity (Figure 8B). In the comparison of beta diversity among the groups, the healthy control group could be differentiated from the other groups on the PCoA plot (p = 0.01), suggesting DSS treatment substantially altered the microbial composition of mice. When the healthy control group was excluded from the comparison, the 2′-FL-High group could be distinguished from the DSS control group, indicating that the microbial composition of the two groups was substantially different (*p* = 0.05). We also observed the same separation between the DSS control and 2′-FL-High group in the estimation of beta diversity between the healthy/DSS control group and the FL-treated groups (*p* = 0.01). We then described the gut microbial composition, which reflects the relative abundance of the detected microbial taxa, and the gut microbial profile of healthy control mice was extremely different from those of mice in the other DSS-treated groups (Figure 8C). However, FL treatment was considered to have reversed the observed DSS-induced alteration because the microbial composition differed between the DSS control group and FL-treated groups. For example, the abundance of *E. coli*, which is considered to be associated with colitis, was increased in the DSS control group (Figure 8C); however, its abundance was lower in all FL treatment groups, especially the 2′-FL-High group, than in the DSS control group (Figure 8D).

## 4. Discussion

The intestine is one of the major organs of the digestive system, and it is divided into the small and large intestine. Between them, the large intestine is mainly responsible for the digestion and excretion of food that was not digested in the small intestine, as well as the absorption of certain electrolytes and water-soluble vitamins. UC is a chronic, recurrent disease of unknown cause that is limited to the large intestine, and it causes inflammation or ulceration of the intestinal wall [29]. The disease is characterized by bloody diarrhea, stool urgency, and abdominal pain with repeated improvement and exacerbation [30]. The prevalence of UC is steadily increasing, but the no curative treatment has been developed [31]. Various new drugs including biologics have been developed, and many clinical studies are in progress. Although these agents can slow the rate of intestinal damage depending on drug responsiveness and clinical response, they are insufficient for the treatment and prevention of chronic disease [32].

At present, the health benefits of HMOs including their prebiotic effect that promotes the growth of intestinal lactic acid bacteria have been established [33]. However, research on the functional and physiological properties of HMOs is limited, especially their effects on intestinal barrier protection, anti-inflammation, and the gut microbiota. In this study, we confirmed that HMOs, namely 2′-FL and 3-FL, obviously improve the symptoms of colitis and reduce inflammation and mucosal damage through alleviating gut microbiota dysbiosis and protecting the intestinal barrier. Furthermore, 2′-FL and 3-FL reverse body weight loss, intestinal shortening, crypt damage, and intestinal permeability induced by DSS treatment. Abnormal oxidative stress and excessive pro-inflammatory cytokine (e.g., IL-6, TNF-α) production are deeply involved in the progress of intestinal inflammation [34]. We recorded remarkable increase in DAI, histopathologic damage, serum TNF-α and IL-6 levels, and MPO activity in mice with DSS-induced colitis. Several studies identified the regulation of cytokines and suppression of inflammatory responses and oxidative stress as pivotal components of treatment strategies for UC [35,36]. The present study revealed that 2′-FL, 3-FL reduced the hydrogen peroxidase-mediated oxidation of halide ions to hypochlorous acid. Furthermore, 2′-FL and 3-FL were revealed to reduce the serum levels of the pro-inflammatory factors IL-6 and TNF-α. Therefore, these data demonstrate that 2′-FL and 3-FL alleviate DSS-induced colitis through anti-inflammatory and anti-oxidant effects. Furthermore, we determined that 2′-FL has stronger effects on MPO activity and serum IL-6 and TNF-α levels than 3-FL, thereby more strongly contributing to reduction in inflammatory symptoms. 

Increasing amount of research indicates that the intestinal mucosal barrier dysfunction induced by chronic inflammation, including reduced gut mucus layer thickness and elevated intestinal permeability associated with a repetitive cycle of intestinal epithelial barrier disruption and inflammation, is associated with the pathogenesis of colitis [37,38]. Ultimately, excessive intestinal epithelial cell apoptosis and TJ proteins dysfunction lead to physical intestinal barrier damage, resulting in increased intestinal permeability [39]. One of colitis treatment strategies is TJ protein, including ZO-1, occludin and claudin, which lead to protection of intestinal mucosa barrier from the infiltration of harmful substances or bacteria by anchoring with actin cytoskeleton and tightly sealing the epithelium [37,40]. In this study, consistent with previous reports, the protein expression of ZO-1 and occludin was significantly decreased by DSS treatment in mice, whereas claudin-2 expression was obviously increased. However, the expression of these proteins was notably reversed by 2′-FL and 3′-FL treatment, suggesting their ability to maintain intestinal integrity. Furthermore, these data were consistent with TJ protein expression in the IL-6-induced cellular barrier dysfunction cell model. These results illustrated that 2′-FL and 3-FL guard against DSS-induced colitis by retaining intestinal barrier function.

Overall, our findings suggest that HMO administration preserves the diversity of intestinal microflora in colitis in a manner that benefits the host. This treatment might contribute to the improvement of intestinal function, although further clinical studies are required for confirmation. The shotgun metagenome sequencing analysis in the study improved the resolution and reliability of the findings compared to the conventional method for predicting intestinal microorganisms based on the 16S rRNA amplicon sequence. Our findings also suggest that 2′-FL administration can improve intestinal function by increasing the alpha diversity of the gut microbiome. Further studies are required to clarify the association between HMO administration and the abundance of *E. coli.*

## 5. Conclusions

In this study, 2′-FL and 3-FL notably alleviated the symptoms of IL-6-induced barrier dysfunction and DSS-induced colitis. 2′-FL and 3-FL administration reverses weight loss, increases DAI, histopathological changes, and changes in inflammatory cytokine levels in mice. These agents additionally improve intestinal barrier function by increasing TJ protein expression. Additionally, 2′-FL and 3′-FL improve the diversity of the intestinal microbial community. Taken together, our results indicate that HMOs including 2′-FL and 3-FL could be applied as functional food ingredients for maintaining intestinal function and preventing colitis.

## Figures and Tables

**Figure 1 nutrients-15-01845-f001:**
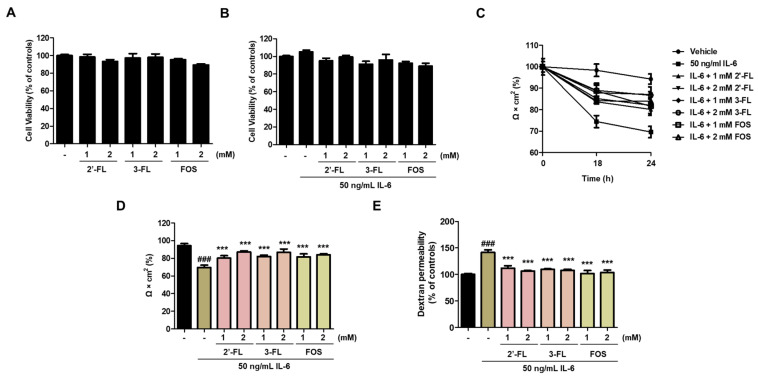
Effects of 2′-fucosyllactose (FL) and 3-FL on the intestinal barrier function of Caco-2 cells. (**A**,**B**) Cell viability after treatment with 2′-FL and 3-FL together with IL-6. (**C**,**D**) Transepithelial electrical resistance (TEER). (**E**) Epithelial paracellular permeability. TEER values were used to analyze intestinal barrier integrity and paracellular permeability, which was determined using a non-resorbable FITC-conjugated dextran probe (FD-4). Cells were pretreated with various concentrations of 2′-FL and 3-FL for 1 h followed by treatment with 50 ng/mL IL-6 for additional 24 h. Cell viability was determined using Cell Counting Kit-8. Cell viability is expressed as a percentage relative to control. Results are presented as the mean ± standard error of the mean of three independent experiments vs. control ### *p* < 0.001; *** *p* < 0.001 vs. IL-6 treatment group.

**Figure 2 nutrients-15-01845-f002:**
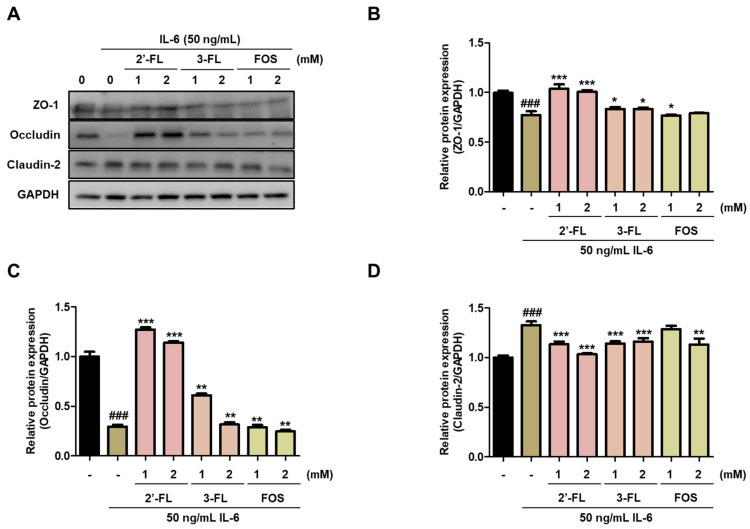
Effects of 2′-fucosyllactose (FL) and 3-FL on tight junction protein expressions. (**A**) Representative expression of zonula occludens (ZO)-1, occludin and claudin-2. Protein expression was measured by Western blot analysis. GAPDH was used as the protein-loading control. Densitometry was performed to quantify the protein expression of (**B**) ZO-1, (**C**) occludin, and (**D**) claudin-2. Results are presented as the mean ± standard error of the mean of three independent experiments vs. control ### *p* < 0.001; * *p* < 0.05, ** *p* < 0.01 and *** *p* < 0.001 versus IL-6-treated group.

**Figure 3 nutrients-15-01845-f003:**
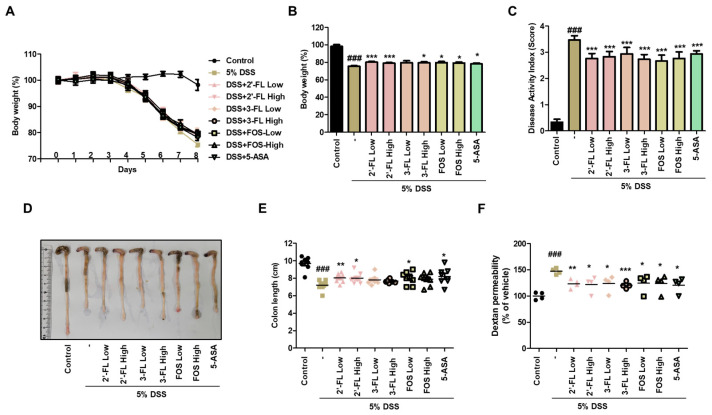
Effects of 2′-fucosyllactose (FL) and 3-FL on body weight, colon length, and serum FITC-dextran permeability in mice with dextran sodium sulfate (DSS)-induced colitis. (**A**,**B**) Body weight. (**C**) Disease activity index. (**D**,**E**) Colon length. (**F**) Serum FITC-dextran permeability. Prior to DSS treatment, mice were orally administered with 2′-FL, 3-FL, fructooligosaccharide (FOS) (1 or 3 g/kg) or 5-aminosalicylic acid (5-ASA, 100 mg/kg). Body weight was monitored prior to administration of 2′-FL, 3-FL, FOS or 5-ASA during the experimental period. The length of the large intestine was measured after being isolated from the sacrificed mouse. Epithelial paracellular permeability was measured using FD-4. Results are expressed as the mean ± standard error of the mean of each mouse in the same group. ### *p* < 0.001 versus control group, * *p* < 0.05, ** *p* < 0.01, *** *p* < 0.001 versus DSS treatment group.

**Figure 4 nutrients-15-01845-f004:**
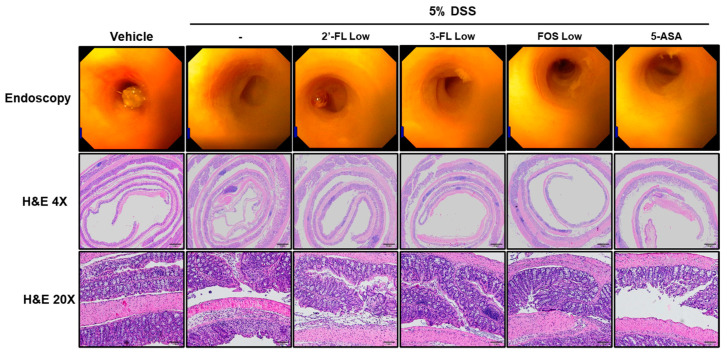
Effects of 2′-fucosyllactose (FL) and 3′-FL on histopathological changes in mice colon tissues of dextran sodium sulfate (DSS)-induced colitis. Endoscopy and hematoxylin- and eosin (H&E)-stained images. DSS-induced mucosal damage evaluated using a mini-endoscope and H&E staining on day 8 of experiment. Represent images are presented. Magnification ×200.

**Figure 5 nutrients-15-01845-f005:**
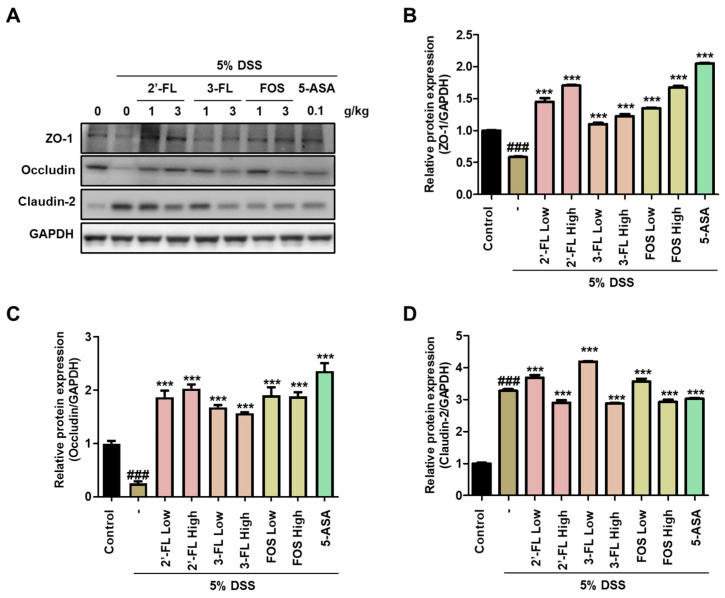
The protective effects of 2′-fucosyllactose (FL) and 3′-FL on tight junctions in mice colon tissue of dextran sodium sulfate (DSS)-induced colitis. (**A**) Representative expression of ZO-zonula occludens (ZO)-1, occludin, and claudin-2 as determined by Western blot analysis. GAPDH was used as the protein-loading control. Representative densitometer (**B**) ZO-1. (**C**) Occludin. (**D**) Claudin-2. The results are presented the mean ± standard error of the mean of three independent experiments. ### *p* < 0.001 versus the control group; *** *p* < 0.001 versus the DSS-treated group.

**Figure 6 nutrients-15-01845-f006:**
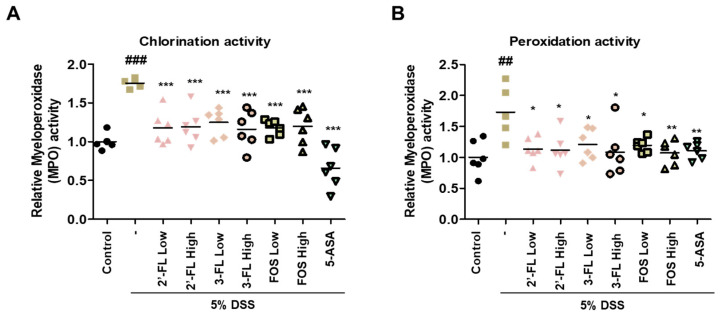
The effects of 2′-fucosyllactose (FL) and 3-FL on myeloperoxidase (MPO) activities in colon tissue of dextran sodium sulfate (DSS)-induced colitis mice. (**A**) Chlorination activities. (**B**) Peroxidation activities. MPO activity was calculated using an MPO activity assay kit. Data are presented as the mean ± SD, ## *p* < 0.01 and ### *p* < 0.001 versus the control group, * *p* < 0.05, ** *p* < 0.01 and *** *p* < 0.001 versus the DSS-treated group.

**Figure 7 nutrients-15-01845-f007:**
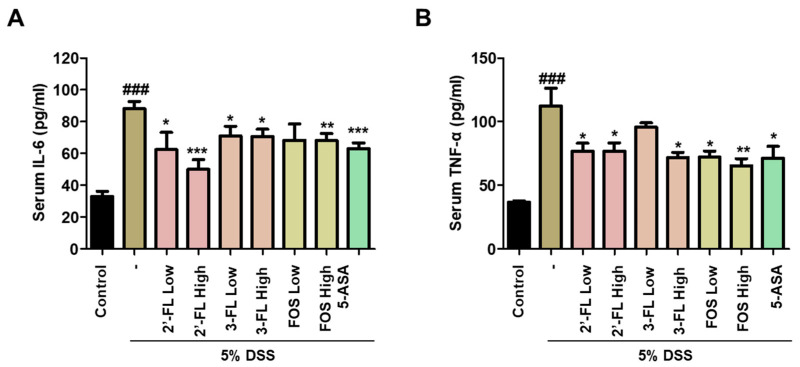
Effects of 2′-fucosyllactose (FL) and 3-FL on the serum levels of pro-inflammatory cytokines of dextran sodium sulfate (DSS)-induced colitis mice. Serum levels of (**A**) interleukin (IL)-6 and (**B**) tumor necrosis factor (TNF)-α as determined by enzyme-linked immunosorbent assay. The results are presented as the mean ± standard error of the mean of each mouse in the same group. ### *p* < 0.001 versus the control group, * *p* < 0.05, ** *p* < 0.01 and *** *p* < 0.001 versus the DSS-treated group.

**Figure 8 nutrients-15-01845-f008:**
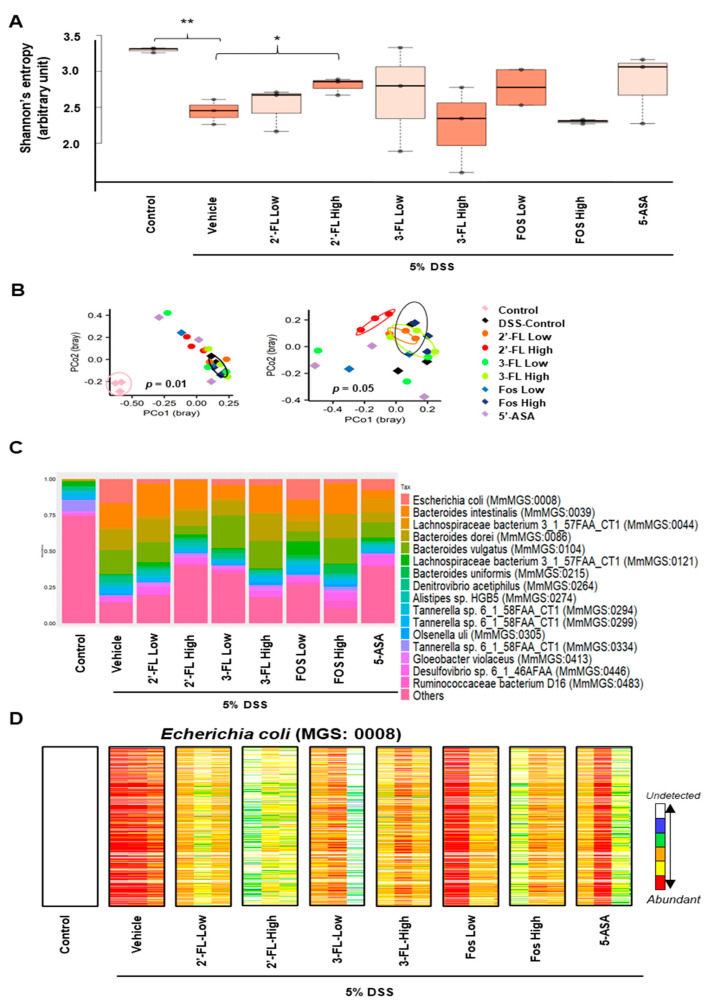
Impact of 2′-fucosyllactose (FL) and 3-FL on the gut microbiome. (**A**) Boxplots presented differential alpha diversity (Shannon’s entropy) among the groups. Boxes represent the inter-quartile range (IQR). The upper whiskers represent the range from minimum (upper quartile − 1.5 IQR) to maximum (lower quartile + 1.5 IQR), and black dots represent outliers excluded from the range. (**B**) Principal coordinate analysis (PCoA) plots presented beta diversity estimated among all groups (left), and all groups but healthy control (right). The distances between samples were determined using Bray-Curtis dissimilarity. (**C**) A stacking bar plot presented the differential core microbial genus composition of the control and HMOs-treated groups. (**D**) Barcode plots showing differentially abundant *Escherichia coli* (*E. coli*; MGS:0008) in the control groups and HMO-administrated groups. Each column indicates individual samples in the groups. Each row indicates the presence of *E. coli* specific gene detected from feces. Abundances of *E. coli* specific genes are indicated by color gradient from white (undetected) to red (most abundant). Each color change represents 4-fold change. Statistical significance was determined using PERMANOVA or the two-tailed *t*-test. * *p* < 0.05, ** *p* < 0.01.

**Table 1 nutrients-15-01845-t001:** DAI scoring system.

DAI Score	Weight Loss (%)	Stool Condition	Rectal Bleeding
0	None	Normal	None
1	1–5		
2	5–10	Loose stools	
3	10–20		
4	>20	Diarrhea	Gross bleeding

## Data Availability

All the data supporting the results were shown in the paper, and can be obtained from the corresponding author.

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
