# Peer review of "2′-Fucosyllactose and 3-Fucosyllactose Alleviates Interleukin-6-Induced Barrier Dysfunction and Dextran Sodium Sulfate-Induced Colitis by Improving Intestinal Barrier Function and Modulating the Intestinal Microbiome"

_nutrients, 2023, doi:10.3390/nu15081845_

Round 1

Reviewer 1 Report

Dear authors,

Thank you for your contribution.  This is a well designed and well executed study.  The manuscript is well prepared and easy to follow.  I have no major corrections or suggestions.  I would just invite you to consider the differences in the mechanisms of the development of colitis between humans and a controlled experiment in an animal model like mice.  The reader needs to understand that while 2'-FL and 3'-FL seem to be very helpful, this conclusion is limited to your animal model where colitis is induced via DSS exposure.  In a human who develops colitis there are many other factors, some known and others unknown.  This needs to be mentioned. 

Author Response

Reviewer 1 comments:

Thank you for your contribution. This is a well designed and well executed study.  The manuscript is well prepared and easy to follow.  I have no major corrections or suggestions.  I would just invite you to consider the differences in the mechanisms of the development of colitis between humans and a controlled experiment in an animal model like mice.  The reader needs to understand that while 2'-FL and 3'-FL seem to be very helpful, this conclusion is limited to your animal model where colitis is induced via DSS exposure.  In a human who develops colitis there are many other factors, some known and others unknown. This needs to be mentioned.

Author answer: We thank you for your thoughtful comments. We agree with your opinion that, in humans who develop colitis, there are many other factors. In this study, we confirmed the effects of HMO on intestinal integrity and permeability using IL-6-induced barrier dysfunction and DSS-induced colitis animal models. Although we cannot verify this under normal intestinal integrity conditions, previous reports indicate that the prebiotic effect of HMOs has been extensively studied. In this study, we found that both 2′-FL and 3-FL decreased intestinal permeability in vivo and in vitro. In other words, these substances selectively reduce inflammation by reducing TJ protein expression after exposure to interleukin (IL)-6 or DSS.

'2'-fucosyllactose is a trisaccharide that is lactose in which the hydroxy group at the 2'-positions (galactose site) has been glycosylated by an α-L-fucose(fucopyranosyl group). Its IUPAC name is Fuc(a1-2)Gal(b1-4)Glc. Whereas 3-fucosyllactose is a trisaccharide that is lactose in which the hydroxy group at position 3 of the glucosyl moiety has undergone formal condensation with the anomeric hydroxy group of fucose(6-deoxy-L-galactose). Its IUPAC name is Fuc(a1-3)[Gal(b1-4)]Glc and SVG image is as follows,

Reviewer 2 Report

The manuscript by Kim et al., has investigated protective effect of human milk oligosaccharides (HMOs) on epithelial barrier function and intestinal inflammation acute dextran sulfate-induced colitis model and IL-6 induced cell model.

In the experiments of DSS induced colitis you are using mouse model, where you want to show beneficial impact of Human milk oligosaccharides. How physiologically explain the impact of mixing human substances with mouse epithelial. I don’t see huge beneficial impact of HMO on the changes in body weight during DSS colitis. How do you explain that? Why the labeling of 2´FL and 3 Fl. Is labeled differently? “ with comma and 3 without comma. This is labeling probably position on the molecule why this difference in writing?

You are everywhere comparing two concentration of individual component, I don’t see significant changes in difference if the concentration used were high or low. This is misleading.

The use of use of methods is very diverse and it is good to use all these methods to characterize the beneficial impact of HMO on the outcome of DSS.  Although I am not completely persuaded by the beneficial  impact of HMO on the outcome of DSS.

Author Response

Reviewer 2 comments:

The manuscript by Kim et al., has investigated protective effect of human milk oligosaccharides (HMOs) on epithelial barrier function and intestinal inflammation acute dextran sulfate-induced colitis model and IL-6 induced cell model.

  1. In the experiments of DSS induced colitis you are using mouse model, where you want to show beneficial impact of Human milk oligosaccharides. How physiologically explain the impact of mixing human substances with mouse epithelial. I don’t see huge beneficial impact of HMO on the changes in body weight during DSS colitis. How do you explain that? Why the labeling of 2´FL and 3 Fl. Is labeled differently? “ with comma and 3 without comma. This is labeling probably position on the molecule why this difference in writing?

Authors answer: We thank you for your thoughtful comments. In this study, we verified the statistical significance of HMO's beneficial impact on changes in body weight during DSS colitis. Also, the effect of HMO was sufficiently supported by histopathology and endoscopy, the expression of tight junction proteins, and inflammatory cytokine levels.

  1. You are everywhere comparing two concentration of individual component, I don’t see significant changes in difference if the concentration used were high or low. This is misleading.

Authors answer: Although the results from the two dosages appear to be equivalent, they are nonetheless regarded as being adequately valuable because even at high concentrations, no side effects or animal toxicity were found.

  1. The use of use of methods is very diverse and it is good to use all these methods to characterize the beneficial impact of HMO on the outcome of DSS. Although I am not completely persuaded by the beneficial impact of HMO on the outcome of DSS.

Authors answer: We agree with your comments. DSS, which has been acknowledged and supported by several research, was used in this investigation to validate the effects of HMOs. In addition, the effects of HMOs were confirmed using histopathological and molecular biological methods. further, we have demonstrated that HMOs can protect the integrity of the intestine in cell study. Therefore, we believe that the effectiveness of HMOs has been verified in various ways.

Round 2

Reviewer 2 Report

no authors reply my comments, although things are discussible i agree it can pass.